# Keep your budget together! Investigating determinants on risky decision-making about losses

**Marc Wyszynski** [1]*, **Adele Diederich** [2]

**1** Department of Psychology & Methods, Jacobs University, Bremen, Germany, **2** Department of Life Sciences & Chemistry, Jacobs University, Bremen, Germany

* m.wyszynski@jacobs-university.de

**Data Availability Statement:** Materials and data are made publicly available on the Open Science Framework (https://osf.io/37myc).

**Funding:** This work was supported by Deutsche Forschungsgemeinschaft grant DFG FOR2104

## Abstract

The present study investigates the influence of framing, different amounts to lose, and probabilities of a risky and sure choice option, time limits, and need on risky decision-making. For a given block of trials, participants were equipped with a personal budget (number of points). On each trial within a block, a specific initial amount is possibly taken from the budget by the outcome of a gamble or the choice of a sure loss option. The goal was to avoid losing points from the budget for not falling below a predefined need threshold. Three different levels of induced need were included. Employing a psychophysical experimental approach, we furthermore tested a sequential component of human risk behavior towards a need threshold inspired by research on animal foraging behavior. Risk-sensitivity models and the Stone-Geary framework serve as generating hypotheses on need thresholds. We found that framing, need, and probabilities influenced risky choices. Time limits and initial amounts moderated the framing effect. No sequential component was observed.

## Introduction

Every day people make multiple decisions, assuming that the choice is based on their "true" preference. However, numerous studies have shown that frames, i. e., the way how a choice problem is described (e. g., [1–3]), may influence the decision-maker's preference. One experimental paradigm for investigating the effect of framing on decision-making ("framing effect" [1]) is the risky choice framing: choice options are framed as gains (e. g., win a lottery with probability $p$; or with probability $p$, X of Z people will be saved) or as losses (e. g., lose a lottery with probability $1 − p$; with probability $1 − p$, $Y = Z − X$ people will die). It has repeatedly been shown that people become more risk-taking in loss-frames than in gain-frames (see e. g., [4–6] for meta-analytic reviews).

The size of the framing effect depends on various variables such as the probability of winning/losing the gamble, the magnitude of outcomes, or problem domains, for instance, simple lotteries or more elaborate settings like the famous Unusual Disease problem [7] (often referred to as "Asian Disease Problem"; however, we use a more contemporary term without labeling the disease with a country or region of origin). Within the latter, even different

("Need-based justice and distributive procedures"), DI 506/13-1, DI 506/13-2. URL: https://gepris.dfg. de/gepris/projekt/259014267. Initials of authors who received each award: AD. The funders had no role in study design, data collection and analysis, decision to publish, or preparation of the manuscript.

**Competing interests:** The authors have declared that no competing interests exist.

diseases may affect the framing effect size [8–10]. Moreover, the probability of winning/losing the gamble and the magnitude of outcomes are factors that can influence risk-taking independently from the framing [8, 11, 12].

Experimental designs and conditions that are not directly related to the choice options may also account for some of the observed differences in the strength of the framing effect. For instance, data obtained from a within-subjects design often show a smaller framing effect size than data obtained from a between-subjects design (see e. g. [4, 10]). Researchers often explain this difference with higher transparency of the framing manipulation in within-subjects studies [13].

Another determinant on risky decision-making is the time available for making the choice. Presenting choice options under various time constraints may produce preference reversals and shifts (e. g., [14–18]) but may also moderate the framing effect. For instance, using a quasi-psychophysical approach (i.e., many data points per condition for each participant) for eliciting data in simple lotteries and applying relatively short deadlines (1s and 3s), Diederich et al. [17] found that time limits can influence risky choice regardless of the frame. In particular, they found that participants are more risk-taking under longer than under shorter deadlines. Furthermore, using the same data collecting approach, Diederich et al. [8, 17], Guo et al. [19], and Roberts et al. [20] found stronger framing effects under shorter time limits than on longer ones. However, Svenson and Benson [18] applied a relatively long deadline for making the choice among lotteries as well as the Unusual Disease problem. Their results showed that a 40-second response deadline reduced framing effects.

In the early 1980s, a few studies in the field of managerial risk behavior have shown that risky decision-making involving gains and losses further depends on the level of aspiration [21, 22]. Defining aspiration levels as reference points or minimum target returns that the decision-maker needs to achieve, the authors found that for 'nonruinous' losses, decision-makers become more risk-taking for outcomes below the aspiration level than for those above the aspiration level. However, managers became less risk-seeking or even risk-averse when ruinous losses (causing "severe liquidity problems for their firms, and possibly bankruptcy"; [22, p. 1242]) were involved. Similarly, self-set goals that challenge the decision-maker in certain ways (e. g., a specific amount of money to earn during an exercise) have been shown to influence risk behavior. In particular, individuals tend to become more risk-taking when they are below their goals or achieving the goal is at risk. [23–25].

Recently, the concept of need, which seems to be related to the ideas of aspiration levels and goals, has been probed as a possible moderator for framing effects. Note that need has multiple meanings between and even within different disciplines that we will not discuss here (see [26] for an overview). In the following, we limit our focus to those definitions/operationalizations of need that have been used in studies with frames and risky choice situations.

Mishra et al. [27] defined need as the "minimum number of individuals to be saved/prevented from dying" in Tversky and Kahneman's [7] Unusual Disease problem. They included two levels of need: low and high. Low need could be met with both the sure and the risky option. High need, however, was unlikely to be met by choosing the sure option. They found that participants tended to choose the risky option more often when need was high. Framing effects, however, were only observed in the control condition, where need was not included. Diederich et al. [8] defined the magnitude of need as the "number of affected individuals who are in need for a treatment" in variations of the Unusual Disease problem. Their findings suggest that decision-makers become less risk-taking in problems concerning a higher number of individuals. They found no relationship between need and framing.

Note that, in both studies, the decision-maker's choices affected other people's need, not their own. When choices affect the need of decision-makers themselves, results are different:

defining need as "the participant's imagined debt", Mishra and colleagues [28] asked participants to choose between two hypothetical investment programs involving different levels of risk, debt (need), and outcome (to pay off the debt). The programs were presented in either a positive or a negative frame. They found that people become riskier when they cannot meet their need by choosing less risky programs. Furthermore, when need was high, the framing effect became stronger; when need was low, the effect disappeared.

Diederich et al. [17] defined (induced) need as an objective lack of resources that are necessary to maintain physiological and mental health and operationalized it as "a preset amount of points participants had to attain" by playing a game for several rounds. The game included one sure and one risky option and was framed either as gain or loss. They found that participants chose the risky option more often when need (points to attain) was high. Furthermore, the effect of need reduced the framing effect because the increase of risky choices was higher in gain-frames than in loss-frames. Using the same definition and operationalization of need, Wyszynski et al. [12] examined the role of it in decisions in which the decision-makers had to take care of both their own need and the need of another person (identified or not). The experimental setting was a charity-like lottery, framed as gain or loss, in which points could be won (self) or lost (other). They found that need mitigates the framing effect.

The framing effect in risky decision-making is often accounted for by prospect theory [1]. To provide a theoretical (normative) account when need is involved, so far two approaches have been considered: 1) Risk-sensitivity foraging theory and 2) the Stone-Geary utility function.

1) Risk-sensitivity foraging theories were originally developed to predict risk-taking in animal foraging behavior. One of the most prominent theories is the so-called "energy budget rule", also known as "z-score model" [29, 30], which, in turn, is an analytical derivation of the "24-h energy budget rule" [31]. The model assumes that an animal, e. g. a small bird, has a need (minimum requirement/lack) of food reserves. To ensure its survival, the bird must meet the need at the end of a certain period (e. g., one day). A period is divided into a certain number of foraging intervals. At each foraging interval, the bird has to decide on how risk-taking it acts. The energy budget rule predicts that animals would become more risk-seeking when less risky actions are unlikely to meet the need. In other words, the more challenging it is to meet the need, the higher is the tendency to take risks. Moreover, Stephens [29] suggested a "sequential component" of risk-sensitivity. That is, a forager whose needs are not yet met might behave more risk-taking when there are only a few foraging intervals left as compared to a forager with many intervals left. Although the energy budget rule was originally developed to predict risk behavior of foraging animals, it has been successfully adopted and generalized to account for human decision-making under risk in the context of need [27, 28, 32–36].

2) Without discussing any details on economic demand theory, which is on the relationship between consumer demand for goods and services and their prices in the market (e. g., [37] in the context of psychological research), need in terms of *minimum* consumption levels for goods and services can be incorporated by the Stone-Geary utility function [38, 39]. It is a functional representation of consumer preferences, and for each good, a utility function is defined, comprising a so-called linear expenditure system. This system can be split into two components (for details see [40]): First, the minimum consumption of each good has to be met. Second, after subtracting the subsistence income that is needed to buy the minimum quantities, the remainder of the consumer's income is spent according to a person's preferences and budget restrictions. Note that the Stone-Geary utility function is defined only for consumption levels exceeding the minimum. Hence, the minimum consumption levels are usually interpreted as indispensable to life. Based on these ideas, Diederich et al. [17] hypothesized that an expected-utility maximizer becomes more risk-seeking with increasing induced need thresholds and regardless of the specific frame.

Thus, both frameworks predict increasing risk-taking behavior with increasing needs but are based on different assumptions and mechanisms. For the energy budget rule, decisions are made based on collecting resources, i. e., starting from no or a low resource level until an upper limit is reached. In contrast, the Stone-Geary utility framework starts with an endowment of resources and expenditure decisions are made until a lower limit is reached.

Bringing the sequential component of risk-sensitivity and Stone-Geary utility framework together, we speculate that a forager would take higher risks at the beginning of a period to ensure meeting the minimum consumption level (need) first. Furthermore, risk-taking behavior would decrease as the amount of resources required to meet the need decreases.

The goal of the present study is twofold: First, we seek to test whether we can observe similar results using an experimental design as in Diederich et al. [17], but with one crucial difference (explained in detail below): Instead of *collecting points* (into a personal account) towards a predefined need threshold, here participants are instructed to *avoid losing points*(from a personal budget) for not falling below a predefined need threshold. That is, need is induced in terms of a minimum requirement of points decision-makers must *keep* in their budget at the end of a predefined number of trials, i. e., one period. In other words, we probe the different tasks (i. e., collecting points into a personal account versus avoiding losing points from a personal budget) as further possible frames that may influence risky choice behavior. Second, we test the concept of the "sequential component of risk-sensitivity" as described by Stephens [29] in the context of human decision-making. According to our knowledge, that has not yet been done.

In summary: Building upon the experimental evidence and theoretical considerations described above, we derive the following hypotheses (main effects): H1: Framing influences choice behavior when making decisions to avoid losing resources from a budget, i. e., gambles are more often chosen in loss-frames than in gain-frames. H2: Amounts initially taken from the budget influence choice behavior; in particular, gambles are less often chosen for larger amounts than for smaller ones. H3: Probabilities of winning/losing a gamble influence choice behavior, i. e., the higher the probability of winning, the more often gambles are chosen. H4: Time limits affect choice behavior, i. e., participants become more risk-taking under longer deadlines than under shorter deadlines. H5: Need influences choice behavior, i. e., the higher the need is, the more often gambles are chosen.

In addition, we hypothesize the following factors to moderate framing effects. H6: Amounts initially taken from the budget in a trial—we expect stronger framing effects for higher amounts. H7: Probabilities of winning a gamble—the size of framing effects increases with probabilities of winning the gamble. H8: Time limits—shorter deadlines enhance the framing effect. H9: Need—we expect weaker framing effects with increasing need thresholds. Finally, we suggest a sequential component of human risk behavior for decisions made to meet a need. H10: During one period (number of preset trials), the number of risky choices decreases as the amount of the resource required to hold the need threshold decreases.

## Experiment

We used a psychophysical data collection approach as used in previous framing studies [8, 11, 12, 17, 19, 20]. That is, instead of presenting few trials to many participants as in a typical social science approach, here fewer participants perform many more trials. The experimental setup and the specific values for choice probabilities, amounts, and deadlines were essentially identical to the one described in Diederich et al. [17]. However, it differed in one crucial respect: Instead of offering points ("you are given x points") at the beginning of each trial (Fig 1, upper row) points are possibly taken ("you are taken x points") from a certain personal budget (Fig 1,

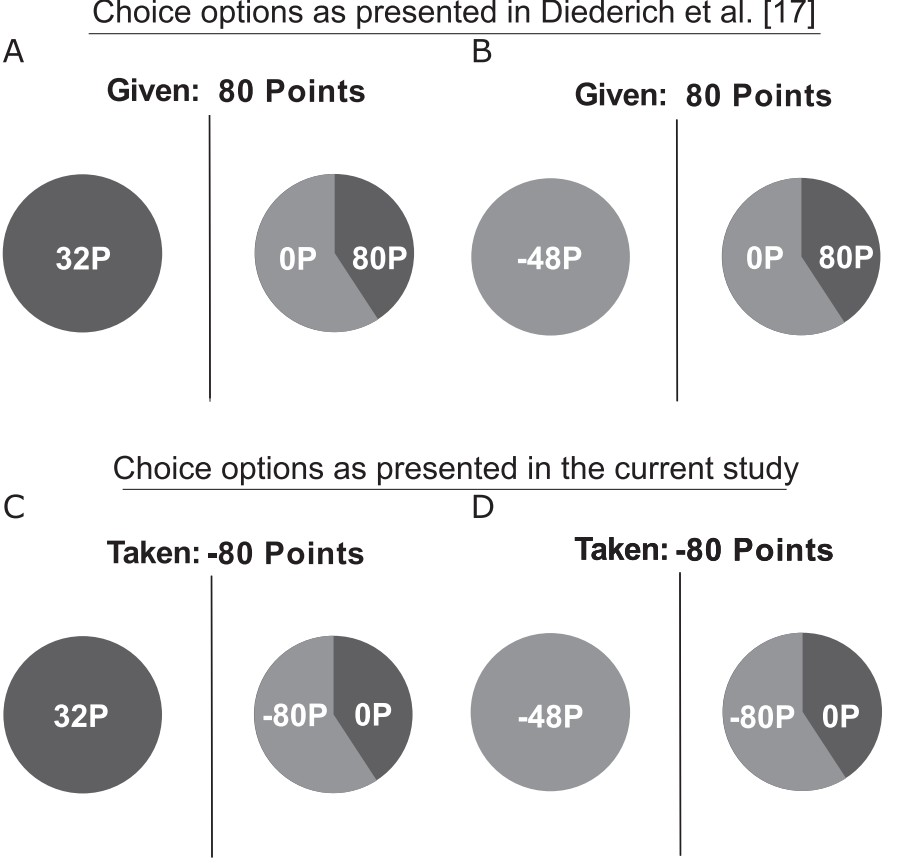

**Fig 1. Gain and loss trial presentations.** Samples from Diederich et al. [17]: "Given: 80 points", "keep 32 P" (gain-frame, panel A), "lose 48 P" (loss-frame, panel B); and from the current study: "Taken: -80 points", "keep 32 P" (gain-frame, panel C), "lose only -48 P" (loss-frame, panel D).

lower row). Choice options were presented in either a gain-frame (Fig 1, left column) or in a loss-frame (Fig 1, right column). All choice situations included one sure and one risky option. Details are presented below.

## Participants

Based on the smallest significant effect size observed in Diederich et al. [17] (Frame×Need; $\beta$-coefficient = .049) and an $\alpha$ of .05, power analysis using the R package "simr" [41] (power calculations are based on Monte Carlo simulations; number of simulations: 500) suggests a sample size of 40 participants to maintain a statistical power of at least 0.9. We included 44 students (18 female, 25 male; between 18 and 36 years old, median = 21) of Jacobs University Bremen in the experiment. All participants were English speakers. They received a 7.50 Euros show-up fee and 0.0025 Euros per point remained of their budgets at the end of the experiment. The experiment had been approved by the Jacobs University Research Ethics Committee. Participants gave their written informed consent before their inclusion in the study. They were screened for their ability to follow the experimental instructions. Each participant was involved in the experiment for approximately 90 minutes.

## Materials and design

The experiment included six blocks, each block consisting of 104 trials. Each trial offered two choice options: one sure option and one gamble. Of the 104 trials, 96 were test trials: 48 with gain-frames and 48 with loss-frames. To assess accuracy and engagement, we also included eight catch trials (4 catch trials per frame).

**Frames.** We manipulated the framing by presenting each trial in a gain-frame and a loss-frame. In the gain-frame, the sure option displays the number of points to *keep* from the amount taken from the personal budget for the current trial. In the loss-frame, the sure option indicates the number of points the participant would *lose only* when choosing the sure option (see Fig 1, panels C and D).

**Amounts initially taken and probabilities.** For the test trials, four different basic point amounts initially taken from the personal budget were selected: −20, −40, −60, and −80, flanked by plus/minus one point amounts, resulting in the following set of four triples: −19, −20, −21; −39, −40, −41; −59, −60, −61; and −79, −80, −81. This was done to minimize the effect of prominent numbers on choice behavior. For evaluation, we collapsed responses to a triple. That is, the three amounts of a triple were treated as one basic point amount in the analysis. The probabilities of winning the gamble were: .3, .4, .6, and .7. The amounts and probabilities were paired to form 48 unique trials. From these pairs, the sure option for each trial was created to match the expected value of the gamble, depending on whether the gamble was framed in terms of a gain or a loss. For instance, in a trial that started by taking 80 points from the budget and a probability of winning the gamble of .4, the sure option would either be "keep 32 points in your budget" in the gain-frame or "lose only 48 points from your budget" in the loss-frame (Fig 1, panel C and D). The catch trials had non-equivalent sure and gamble options. In half of the catch trials, the sure option had a higher expected value (initial amount × .9) than the gamble option (probability of winning = .1). In the other half, the gamble option had a higher expected value (probability of winning = .9) than the sure option (initial amount × .1).

**Need.** At the beginning of each block of 104 trials (one period), the personal budget was set to 5,600 points. We included three need levels: 2,800 and 3,600 points, and 0 points as control. One need level was fixed within one given block of trials. Note that the need thresholds are the same as in Diederich et al. [17]. However, instead of accumulating the *given* points to meet these thresholds (starting from zero), here, for each trial, points are *taken* from the personal budget of 5,600 points, and the goal was not to fall below them. If the participant fell below the required minimum score, they received no payoff for that block.

**Time limits.** We included two response time limits: 1 second or 3 seconds for a given block of trials. The exact values of time limits were adopted from previous studies investigating the effect of time limits on risky choice using similar designs [8, 17, 19]. Note that the long time limit of 3 seconds has been determined by adding one standard deviation to the mean response time observed in pilot studies not including any time limits.

Altogether we have 12 amount levels (collapsed to 4 in the evaluation), 4 probabilities of winning/losing, 2 frames, 3 need levels, and 2 time limits. Amounts multiplied by probabilities result in 48 test trials. Adding the 4 catch trials and including all experimental conditions results in a total of 624 trials (52 × 2 × 3 × 2) per participant (within-subjects design). A given time limit and a given need level were fixed within one block of 104 trials.

## Apparatus

Stimulus presentation and response registration were controlled by one of six computer systems (hardware specifications are found in S1 Appendix).

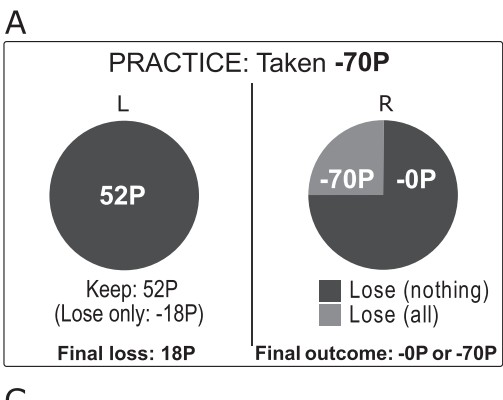

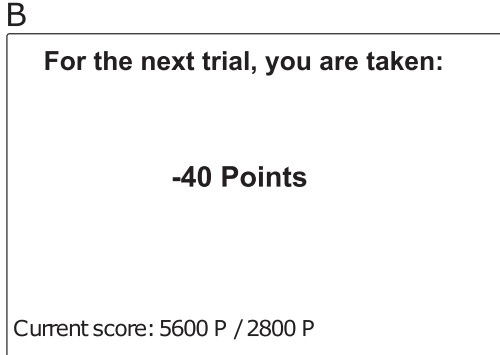

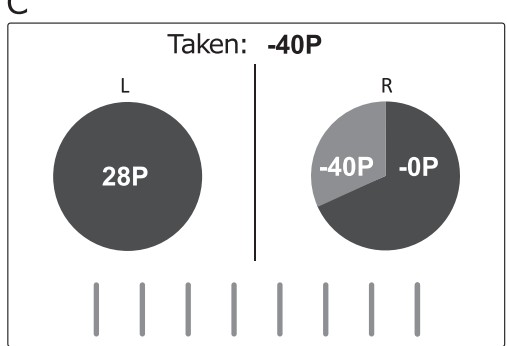

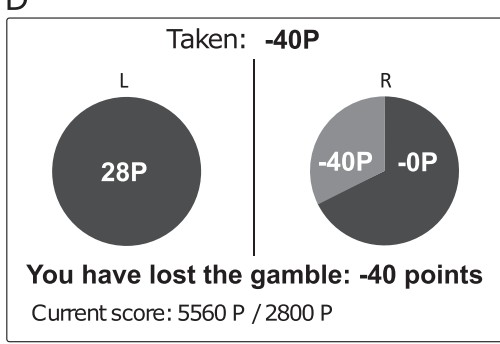

**Fig 2. Example of a guided practice trial (panel A) and timeline for one trial in a gain-frame (panel B-D).** The screen displaying the amount of points initially taken was presented for 2.5s (panel B). It also displayed the induced need for the block of trials and the remaining budget. The screen displaying the choice options was presented for either 1s or 3s, depending on the experimental condition (panel C). The bars below the pie-charts indicate the available time for making a choice (speed by which the bars were removed). The feedback screen (panel D) announced the result of the decision, the remaining budget, and the need target. It was presented for 2.5s.

The monitor display had a white background. Choice options were represented as two pie charts displayed next to each other in the center of the screen. An example of the display is shown in Fig 1 (panel C, gain-frame; and panel D, loss-frame). The sure or the gamble option was randomly presented on the left or the right side. Available time for making a decision was indicated by eight vertical bars displayed at the bottom of the screen. The bars disappeared one by one as a function of the preset time limits, i. e., the given time limit divided by eight (see Fig 2). Note that participants knew the total number of trials per block (i. e., the length of one period). However, they received no information about the current trial number, i. e., they had to count the trials themselves or estimate the number of the remaining trials. See S1 Appendix, Brainard, and Kleiner et al. [42, 43] for more details.

## Procedure

Participants came to the 'Laboratory for Behavioral and Social Sciences' of Jacobs University Bremen. The lab provides various facilities allowing us to test up to six participants simultaneously. The experimenter asked the participants to read the instructions carefully and explain the experiment in their own words. They then performed a practice session consisting of 12 trials (six gain-framed and six loss-framed trials). During practice, they were first guided through the first six trials by receiving precise instructions on what to do (e. g.,

"chose the gamble" or "chose the sure option") followed by an explanation of the consequences of the corresponding choice. After the guided practice, participants completed additional six practice trials where they could respond freely. In all of the guided practice trials and the first two practice trials, a legend appeared below the pie charts for each option. The legend explains the amounts that could be won or lost (see Fig 2, panel A). This legend faded away step by step during the practice. No time limits and no need points were applied in practice trials.

The first two experimental blocks included the zero need condition with a 3s-time limit in the first block and a 1s-time limit in the second block for all participants. These blocks served as a benchmark. For the remaining four blocks, the two need conditions (2,800, 3,600) were paired with the two time limit conditions (1s; 3s). The pairs were pseudo-randomized across participants with the same need condition presented in two consecutive blocks. Within each block, the 104 trials were presented in random order. At the end of each block, participants had a break. They could continue when they were ready for the next block. At the end of the experiment, one of the six blocks was selected randomly for payoff.

The experimental trials started by showing the amount of points initially taken from the budget for that trial, the budget, and the need (since the way how the amount initially taken is displayed might be misinterpreted as double negation, participants were given clear instructions for proper comprehension; recall that we checked comprehension of the task by asking participants to explain the experiment in their own words). This screen was displayed for 2.5s (Fig 2, panel B). The subsequent screen showed the choice options (sure option and gamble) and a visual timer indicating the time limit for that particular trial (Fig 2, panel C). The screen lasted for 1s or 3s, depending on the experimental condition (time limit). Participants had to respond within the time limit; otherwise, they have lost the amount indicated before each trial (Fig 2, panel B). The last screen (Fig 2, panel D) provided feedback about the outcome of the decision. The feedback depended on whether the participant chose the gamble ("you have won/lost the gamble") or the sure option ("you chose the sure option"). In addition, the feedback screen shows the points remaining in the budget after subtracting the outcome. It lasted for 2.5s. After the offset of the screen, the next trial started.

## Statistical methods

For data evaluation, we used descriptive statistics and regression analysis. We analyzed choice frequencies with generalized linear mixed-models (GLMM), fitted with the Laplace approximation (we use GLMMs since they have been shown to be more flexible, accurate, powerful, and suited for categorical data analysis than a repeated measures ANOVA which has been a common approach in the past [44, 45]). Specifically, we assume an underlying binomial location-scale distribution (logit model). We performed two regression models. For testing hypotheses 1 to 5, i. e., the main effects, we included the following factors as explanatory variables (categories in parentheses; first categories served as reference): Frame (loss; gain), Time limit (1s; 3s), Need (0; 2,800; 3,600), Probability, abbreviated "Prob" (.3; .4; .6; .7), Initial Amount Taken, abbreviated by "IAmT" (−20; −40; −60; −80). To reduce the type I error rate, we included "participants" and "trial numbers" (within one block) as random effects [46]. For testing hypotheses 6 to 9, we performed a 2-way-interaction effects model. Note that we removed any impact of a sequential effect by including the trial number as a random factor in both models. Finally, we tested H10 exploratory on the individual level and we performed Rosner's test to detect outliers and two-sided two-samples t-tests to compare proportions of risky choices between groups of participants. We used the computing environment R (version 4.0.1; packages: 'blme', 'lme4', 'descr', 'psych', 'EnvStats'; [47–52]) for carrying out the analyses.

## Results

Excluding the data from one participant due to a high error rate for the catch trials (>50%), we analyzed the data of the remaining 43 participants. The average proportion of catch trials answered correctly was 76.3%; catch trials were removed for the descriptive analysis and treated as missing values in the inference analysis. Of the remaining 24,768 test trials, 668 were timeouts or anticipatory responses (response time < .01s) and were also removed from further analysis.

In 60% of the valid trials, participants chose the gamble. On average, they kept from their initial 5,600 points budget 2,953 points for need level 0; 2,998 points for need level 2,800; and 2,994 points for need level 3,600. For need level 2,800, participants fall below the threshold in 10.5% of the blocks, and for need level 3,600, in 98.8% of the blocks.

Mean response times among 3s (1s) time condition for gain-framed trials were 1.33s (.65s) and for loss-framed trials 1.31s (.65s) with the standard deviations of .58 (.15) and .59 (.16), respectively. Further analysis of the response times is not part of the current investigation.

The generalized linear mixed-models analysis showed statistically significant main effects for Frame, Probability, and Need (Table 1, main effects model). In particular, the risky option was chosen significantly more often in the loss-frame 65.5% than in the gain-frame (55.0%; supporting H1). The higher the probabilities of winning the gamble, the more often it was

**Table 1. Generalized linear mixed-models.**

| | Main effects model | | | | Interactions model | | | |
|---|---|---|---|---|---|---|---|---|
| | Est. | SE | t-value | p-value | Est. | SE | t-value | p-value |
| Frame (gain) | −.525 | .029 | −18.054 | <.001 | −.538 | .090 | −5.974 | <.001 |
| IAmT (−40) | −.078 | .041 | −1.909 | .056 | −.102 | .059 | −1.727 | .084 |
| IAmT (−60) | −.040 | .041 | −.986 | .324 | .028 | .059 | .474 | .636 |
| IAmT (−80) | −.043 | .041 | −1.039 | .299 | .046 | .059 | .769 | .442 |
| Probability (.4) | .348 | .039 | 8.969 | <.001 | .389 | .054 | 7.151 | <.001 |
| Probability (.6) | 1.478 | .041 | 35.723 | <.001 | 1.559 | .060 | 25.982 | <.001 |
| Probability (.7) | 1.617 | .042 | 38.389 | <.001 | 1.682 | .061 | 27.482 | <.001 |
| Time limit (3s) | −.010 | .029 | −.362 | .717 | −.152 | .042 | −3.626 | <.001 |
| Need 2800 | .089 | .035 | 2.516 | .012 | .082 | .051 | 1.597 | .110 |
| Need 3600 | .231 | .035 | 6.522 | <.001 | .202 | .051 | 3.936 | <.001 |
| Frame× IAmT (−40) | | | | | .046 | .082 | .561 | .575 |
| Frame× IAmT (−60) | | | | | −.131 | .082 | −1.595 | .111 |
| Frame× IAmT (−80) | | | | | −.169 | .082 | −2.056 | .040 |
| Frame× Prob. (.4) | | | | | −.084 | .078 | −1.081 | .280 |
| Frame× Prob. (.6) | | | | | −.155 | .082 | −1.877 | .061 |
| Frame× Prob. (.7) | | | | | −.124 | .084 | −1.481 | .139 |
| Frame× Time limit | | | | | .271 | .058 | 4.673 | <.001 |
| Frame× Need 2800 | | | | | .014 | .071 | .203 | .839 |
| Frame× Need 3600 | | | | | .056 | .071 | .794 | .427 |
| (Intercept) | −.149 | .113 | −1.323 | .186 | −.140 | .121 | −1.155 | .248 |

*Note*. Generalized linear mixed-models fit by maximum likelihood (Laplace Approximation). Dependent variable: Responses (0 = sure; 1 = risky option). Main effect model includes Frame, IAmT, Probability, Time limit, and Need as independent variables. Interaction model includes main effects and hypothesized 2-way-interactions. Reference categories: Frame (loss), IAmT (−20), Probability (.3), Time limit (1s), Need (0). Number of observations: 24,100. Random effects in main effects model: Trials, 104 (random intercept variance: .002) and participants, 43 (random intercept variance: .452). Random effects in interactions model: Trials, 104 (random intercept variance: .002) and participants, 43 (random intercept variance: .453).

chosen (.3: 41.9%, .4: 49.7%, .6: 73.4%, and .7: 75.8%; supporting H3). When need thresholds are included, participants chose the risky option more often as compared to the no-need condition, i. e., the baseline (Need 0: 58.1%, Need 2,800: 59.9%, Need 3,600: 62.6%; supporting H5). H2 and H4, that is, IAmT and time limits affect choice behavior, respectively, was not supported by our data.

As outlined in the introduction, we tested several factors as moderators of the framing effect (IAmT, Probability, Time limits, and Need). A 2-way interaction analysis (Table 1, interactions model) showed statistically significant interaction effects of Frame by IAmT and Frame by Time limit but not for Frame by Probability and Frame by Need. In particular, we observed a significantly stronger framing effect for high IAmT (−80; 53.9% chose the gamble in the gain-frame and 66.5% in the loss-frame) than for low IAmT (−20; 56.4% chose the gamble in the gain-frame and 65.6% in the loss-frame). Furthermore, we found a stronger framing effect for the shorter deadline (53.8% chose the gamble in the gain-frame and 67.0% in the loss-frame) than for the longer deadline (56.2% chose the gamble in the gain-frame and 64.0% in the loss-frame).

Hypothesis 10 is concerned with a possible sequential impact of need on choice behavior. We compare the present results with those of Diederich et al. [17]. Note that the following test was not performed in the previous study. To test H10, we partitioned the number of trials of each participant in two groups (because we did not find any research on suitable alternatives, we simply performed the group partitioning at 50% of the need): Group $G_{\leq 50\%}$ includes those trials halfway to the need threshold, i. e., trials in which the participant has lost (current study) or collected (study of Diederich et al. [17]) up to 50% of the need (i. e., lost/collected 0 to 1,400 points in the Need 2,800 condition; and lost 0 to 1,000 of 5,600 points or collected 0 to 1,800 points in the Need 3,600 condition, respectively); and Group $G_{>50\%}$ includes those trials in which the participant has lost/collected more than 50% and up to 100% of the need (i. e., lost/collected 1,401 to 2,800 points in the Need 2,800 condition; and lost 1,001 to 2,000 of 5,600 points or collected 1,801 to 3,600 points in the Need 3,600 condition, respectively). For the current study, we expect more risky choices in $G_{\leq 50\%}$ than in $G_{>50\%}$ as hypothesized in H10. For the study of Diederich et al. [17], we expect the opposite effect, i. e., more risky choices in $G_{>50\%}$ than in $G_{\leq 50\%}$ as suggested by the sequential component of risk-sensitivity in the context of collecting resources.

Fig 3 shows the proportion of choosing the gamble in $G_{>50\%}$ as a function of the proportion of choosing the gamble in $G_{\leq 50\%}$ for each participant and need condition. The left panel refers to "decisions about losses" (the current study); the right panel refers to "decisions about gains" (study by Diederich et al. [17]). Each choice proportion (one symbol) is based on 96 trials. Choice proportions in the upper left and lower right quadrant indicate that choice behavior changed from choosing the gamble less (more) often before half of the need had been obtained and more (less) often afterward. Off-diagonal choice proportions in the upper right and lower left quadrants indicate a shift from more to less (or less to more) risky choices. The Rosner's test detected one outlier in the data of the study on decisions about gains. In $G_{>50\%}$ of the 2,800 need condition, gambling proportions of one participant differed significantly from the other observations (right panel, lower left quadrants). We removed the outlier for the following analysis.

To give a statistical account of the observed change in choice behavior, we performed a t-test for testing the difference between the choice proportions in $G_{\leq 50\%}$ and $G_{>50\%}$, separate for each participant and need level. For the present study (decisions about losses), about 17% of the participants changed their choice behavior (more or less risky choices before or after they reached half of the need level) for need level 2,800 and 16% for need level 3,600. For the former study (decisions about gains), about 13% of the participants changed their choice behavior for

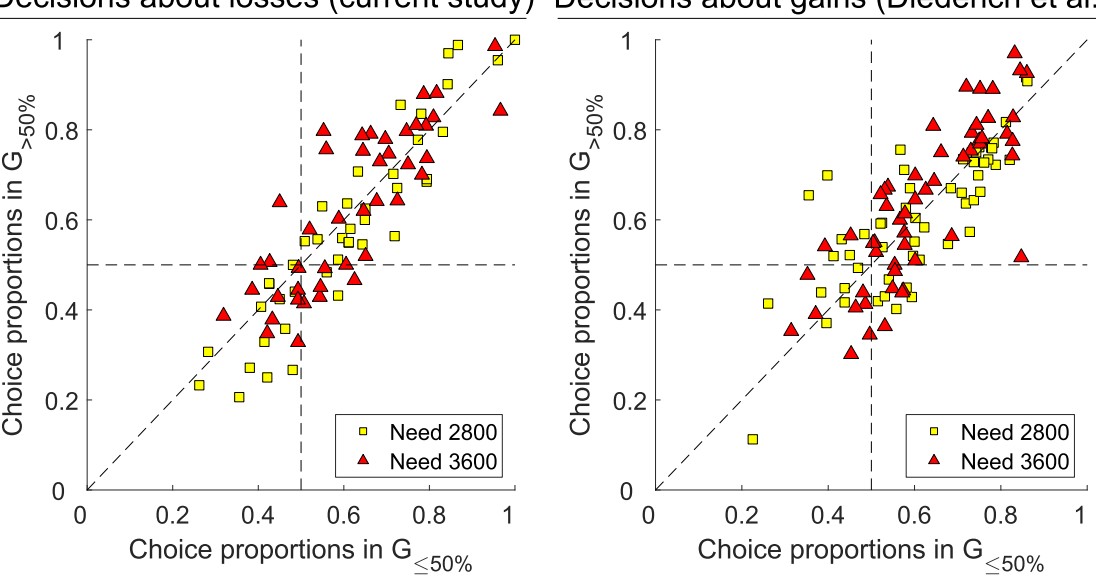

**Fig 3. Proportions of choosing the gamble in $G_{>50\%}$ as a function of the proportion of choosing the gamble in $G_{\leq 50\%}$ for each participant.** Left: decisions about losses (current experiment). Right: decisions about gains (study by Diederich et al. [17]). The x-axis displays choice proportions in trials in which the participant has lost/collected 0 to 1,400 points in the Need 2,800 condition; and lost 0 to 1,000 of 5,600 points (left) or collected 0 to 1,800 points (right) in the Need 3,600 condition, respectively (0 to 50% of need). The y-axis displays choice proportions in trials in which the participant has lost/collected 1,401 to 2,800 points in the Need 2,800 condition; and lost 1,001 to 2,000 of 5,600 points or collected 1,801 to 3,600 points in the Need 3,600 condition, respectively (50 to 100% of need). Squares correspond to Need 2,800 and triangles to Need 3,600 condition.

need level 2,800 and 15% for need level 3,600. Furthermore, we did not find any difference between the two studies with respect to a given need level. Details about the statistics are found in S2 Appendix.

## Summary and discussion

Previous work has demonstrated that the framing of a choice option as gain or loss, amounts and probabilities describing the choice option, and time constraints for making a choice affect decision-making under risk [11, 17, 19]. Moreover, it has been shown that induced need (i. e., a minimum requirement that has to be met) affects the decision-maker's risk-taking behavior [17].

In the current study, we investigated whether we can find similar effects when modifying the design as used by Diederich et al. [17]. Instead of *collecting* points (into a personal account) towards a predefined need threshold, participants of the present study were instructed to avoid *losing* points (from a personal budget) for not falling below a predefined need threshold.

The results supported most of our hypotheses: We observed that framing influenced choice behavior. The risky option was chosen more often in loss-frames than in gain-frames mainly corroborating the results of previous studies on risky choice framing effects once more (see e. g., [4–6, 9, 53]).

Based on previous research suggesting higher risk-taking when the decision involved smaller than larger amounts [8, 11, 54], we hypothesized that individuals will choose the risky option less often when a large number of points is initially taken from the budget as compared to a small number. However, our results do not indicate an effect of different amounts on risk-taking. Thus, we found no support for our second hypothesis.

The frequency of risky choices increased with increasing probabilities of winning the gamble. This finding supports previous research investigating the role of probabilities in risky choice framing paradigms [8, 12, 17, 19]. However, despite the similar designs, it contradicts the results of De Martino et al. [11] and Sundelin et al. [55] who found less risky choices for higher than for lower probabilities.

The data could not support our hypothesis on increasing numbers of risky choices with decreasing time limits. The impact of time constraints on risky choice seems to depend on various factors. Previous studies suggested that the offered time limits themselves seem to be crucial: Under very short time limits (e. g., $\leq$ 1.5s), the effect appears more often as compared to longer but still relatively short constraints (e. g., 3s or 6s [17, 56]). When using longer time limits (e. g., 8s vs. 32s) risk-taking seem to decrease under shorter time limits [57, 58]. Other studies did not find any main effects of time limits on risky choice behavior at all [8, 12, 19]. Furthermore, a time limit effect may also depend on individual differences [17].

As expected, need influences the number of risky choices. Participants chose the risky option more often when need thresholds were included. That is, need seems to enhance risk-taking behavior suggesting that people in need may take higher risks when spending amounts from a budget as compared to people in no need. The finding corroborates previous work on need in risky decision-making in which need was operationalized differently, i. e., as a minimum requirement of a resource that has to be met by collecting it (e. g., [17, 27, 28]). Indeed, the present results show even stronger evidence for the aforesaid relation between risk-taking behavior and need than Diederich et al. [17] observed. Here the proportion of risky choices was higher for both need levels, i. e., the lower and the higher, than for the zero need level. In the previous study, the relation was only found for the higher need level. It is possible that maintaining/obtaining a preset need threshold serves as additional framing. Further research is needed here.

Furthermore, we investigated several factors as moderators of framing effects: initial amounts taken from the budget, probabilities of winning the gamble, time limits, and need. We observed stronger framing effects for high amounts than for low amounts, as expected. This finding is in line with previous studies suggesting that losses loom larger than gains for higher amounts of money, but not for smaller amounts [59, 60]. This effect seems to be stable across different types of outcomes, e. g., it also occurs when the outcome of risky choice affects the number of human lives to be saved (hypothetically) instead of gaining money [8]. With the present study, we demonstrate that different amounts modify the framing effect also for decisions about losses.

Our findings do not support previous research that demonstrated the probabilities of winning (or losing) a gamble moderating the framing effect [9, 11, 17, 61, 62]. Based on those studies, we expected the framing effect to become stronger with increasing probabilities of winning the gamble. In the current study, however, we did not observe any relationship between probabilities and the size of the framing effect. The different design of the present study (i. e., participants made decisions about losses even in the gain-frame) and individual differences as found in Diederich et al. [17] may be reasons for the divergent findings.

As hypothesized, we found that shorter time limits enhanced the framing effect. That is, under shorter time limits, participants become more risk-taking in loss-frames and less risk-taking in gain-frames than under longer time limits. The findings support previous studies that have provided similar results [8, 17, 19, 20, 63]. Other studies, however, indicate either no relationship of time limits and the framing effect [12] or they found the opposite effect as in the present study, i. e., weaker framing effects under shorter time limits [18, 64]. Note that some researchers used much longer constraints for the short time limit condition (up to 60s [18, 63, 64]) as compared to the current study (1 second). Our results are in line with a dual-

process approach, where the intuitive system precedes the deliberative one (see e. g., [65, 66]). That is, the deliberative is considered to a less extent than the intuitive when the time to respond is constrained to short limits. With longer time limits, the influence of the intuitive system wears off, the deliberative system drives the decision process and by that, the framing effects diminish (for details see [8, 19, 67]).

Based on previous work [12, 17, 27], we expected framing effects to occur less often with increasing need thresholds. Our current findings provide no support for this hypothesis. The basic design was similar to the one used by Diederich et al. [17], but they differ with respect to the additional framing in terms of maintaining/obtaining a need threshold. This may be an explanation for the different results. More research is needed here.

Inspired by two approaches that include need in a broader sense—risk-sensitivity foraging theory and the Stone-Geary utility function—we investigated whether choice behavior changes depending on the need level and when it is half reached during one period. Both frameworks predict increasing risk-taking behavior with increasing needs, but are based on different assumptions and mechanisms. For the risk-sensitivity foraging theory, decisions are made based on collecting resources, i. e., starting from no or a low resource level until an upper limit is reached (energy budget rule). In contrast, the Stone-Geary utility framework starts with an endowment of resources and expenditure decisions are made until a lower limit is reached. The effect of need on risky choices has previously been investigated in situations as described for the energy budget rule (e. g., [17, 27, 28]), the present study, however, was concerned with a situation that resembles the Stone-Geary utility framework. Our findings are in line with previous work suggesting increasing risk-taking with increasing needs as proposed by both frameworks.

Furthermore, the "sequential component" of risk-sensitivity was originally considered for animal foraging behavior, i. e., assuming that an animal has a limited number of foraging intervals within a period (e. g., one day). The number of remaining intervals may influence its risk behavior when collecting forage [29]. However, it has not yet been examined in the context of human decision-making. Under consideration of the Stone-Geary utility framework, we hypothesized the decision-maker to become more risk-taking at the beginning of a period to ensure meeting the need first. In the course of a period, we expected risk-taking to decrease as the number of points required to meet the need decreases (note that, using a psychophysical data collection approach, the design of our experiment allowed us to observe the change in risk behavior between early and later intervals within a period). The data do not support our hypothesis. We found only a relatively small number of participants who changed their choice behavior after half of the need level was reached (17% for need level 2,800 and 16% for 3,600), and only some of them became less risk-seeking. We further analyzed data from the study of Diederich et al. [17] that examined decisions as described in the risk-sensitivity foraging theory to compare it with the current results. In the setup of Diederich et al. [17], one would expect that a decision-maker whose needs are not yet met becomes more risk-taking with a decreasing number of remaining intervals. Similar to the present results, 13% (for need level 2,800) and 15% (for 3,600) of the participants changed their behavior. Consequently, our findings revealed no clear indications of a sequential component, as described by Stephens [29] for animal foraging behavior, in human decision-making under risk, neither for decisions about losses (current study) nor for decisions about gains (study of Diederich et al. [17]). Note that human risk behavior may be different in decisions about food than in decisions about monetary amounts [68]; thus, our study can not provide any information about a sequential component in human foraging behavior.

To conclude: The current study aimed to probe the effect of a "hyper-frame" in comparing it with a study by Diederich et al. [17]. In their study, a risky choice paradigm (a choice

between a lottery and a sure option, framed either as gain or loss) served to *obtain* a (need) threshold by providing an amount *given* at the beginning of each trial. The current study used an identical design except that now the game served to *maintain* a threshold by reducing an amount *taken* at the beginning of each trial from a personal endowment. The main effects with respect to framing, need threshold, probabilities, and time limits as moderator for the framing effect are basically the same in both studies. However, in contrast to Diederich et al. [17], the size of the framing effect in the current study was not affected by the need threshold. That is, we could observe a hyper-frame effect.

## Supporting information

**S1 File. Data availability.** Materials and data are made publicly available on the Open Science Framework (https://osf.io/37myc).
(TXT)

**S1 Appendix. Apparatus in detail.**
(PDF)

**S2 Appendix. T-tests: Preference changes.**
(PDF)

## Author Contributions

**Data curation:** Marc Wyszynski.

**Formal analysis:** Marc Wyszynski, Adele Diederich.

**Funding acquisition:** Adele Diederich.

**Investigation:** Marc Wyszynski.

**Methodology:** Marc Wyszynski, Adele Diederich.

**Project administration:** Marc Wyszynski.

**Resources:** Marc Wyszynski, Adele Diederich.

**Software:** Marc Wyszynski.

**Supervision:** Adele Diederich.

**Validation:** Adele Diederich.

**Visualization:** Marc Wyszynski, Adele Diederich.

**Writing – original draft:** Marc Wyszynski, Adele Diederich.

**Writing – review & editing:** Marc Wyszynski, Adele Diederich.

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
