## [Decision Letter · Decision Letter 0]

1 Dec 2021

PONE-D-21-23927Keep your budget together! Investigating determinants on risky decision-making about losses.PLOS ONE

Dear Dr. Wyszynski,

Thank you for submitting your manuscript to PLOS ONE. After careful consideration, we feel that it has merit but does not fully meet PLOS ONE’s publication criteria as it currently stands. Therefore, we invite you to submit a revised version of the manuscript that addresses the points raised during the review process.

I sent your manuscript to two experts in the field.  Both reviewers list a number of major issues that would need to be addressed before this paper is suitable for publication.  I am somewhat inclined to reject this paper, however I will offer you the opportunity to revise the manuscript if you think you can adequately address the reviewers' concerns.  One thing that I will specifically require for a revision is either an increase in the sample size, or a replication study.  Based on the reviews and my reading of the manuscript, I don't think it will be possible to draw meaningful conclusions from the data without additional data being collected.  If you don't think you can satisfactorily address the comments made by the reviewers, then you should probably consider submitting your paper to another journal.  If you do decide to revise the paper then I will examine it first, and then send it back to the original reviewers if I feel that it has been suitable revised for re-review.

We look forward to receiving your revised manuscript.

Kind regards,

Darrell A. Worthy, Ph.D

Academic Editor

PLOS ONE

https://journals.plos.org/plosone/s/file?id=ba62/PLOSOne_formatting_sample_title_authors_affiliations.pdf"

Reviewers' comments:

Reviewer's Responses to Questions

**Comments to the Author**

1. Is the manuscript technically sound, and do the data support the conclusions?

Reviewer #1: Partly

Reviewer #2: Yes

2. Has the statistical analysis been performed appropriately and rigorously? 

Reviewer #1: No

Reviewer #2: Yes

3. Have the authors made all data underlying the findings in their manuscript fully available?

Reviewer #1: Yes

Reviewer #2: Yes

4. Is the manuscript presented in an intelligible fashion and written in standard English?

Reviewer #1: Yes

Reviewer #2: Yes

5. Review Comments to the Author

Reviewer #1: In this study, student subjects are asked to make very many choices between tiny gambles and their expected values. Standard theory predicts that they are indifferent and so whatever they do, is consistent with it.

The time limits are very strict. As a result, in “catch” trials, the dramatically less attractive option was chosen as often as 25% of the time. More generally, it is not clear why we should be interested in time limits here. If this is an attempt to test Stephens’ model, it seems misguided because we only have one interval in the experiment (only it’s longer or shorter).

As far as I can tell, the authors run regressions treating each subject-trial as one observation. Observations coming from the same individual are obviously not independent. Strictly speaking, there are thus only 27 independent observations. For example, at least 20 out of 27 subjects would have to make more risk-seeking choices in on framing than the other for the p value to be below 0.5.

My understanding is that the “needs” were payoff-irrelevant. How were they induced? I don’t know because the instructions are not provided.

Thus, generally speaking,

1 the choice of experimental variables is in my opinion poorly motivated,

2. the design of the study leaves limited space for meaningful inference

3. the study is grossly underpowered.

Minor issues:

p. 2 “Unusual disease” – It’s customary to refer to it as “Asian disease” instead. Is politically incorrect now? I can’t see how the use of this term implies anything wrong about the Asians. I understand the reasons not to name real diseases after geographic locations https://www.nature.com/articles/s41588-020-0617-2 but none of them applies to an abstract, fictitious Asian Disease considered by T&K.

p. 6 “80 points were initial taken from the budget” -- ? should it be “initially”? anyway, I am not sure it’s the optimal term. It suggests that even more point are taken from the budget subsequently (in the same trial) which is not the case.

Figure 1 is confusing. Why do panels A and B have the same label? Same for panels C and D.

Reviewer #2: This study built off of previous gain-framed research (Diederich et al., 2020) to examine the interaction between framing effects, initial loss magnitude, probability, time limits, and need/a goal amount of points in loss contexts. Results showed that framing, probability, and need/goals influenced risky choice: individuals chose the riskier option more in loss-frames, when the initial loss magnitude was small, when the probability was high, and when need/a goal was included. Results also showed that shorter limits magnified the framing effect. This work has potential to advance our understanding of factors that influence risky decision-making, but there are some concerns on the framing of the Introduction, sample size, and some of the methodology.

Major:

(1) Most of the introduction is clear, and most of the hypotheses are clearly and logically generated from the reviewed literature. However, I am somewhat hesitant about using foraging behavior and the Stone-Geary utility framework to motivate some aspects of the study and H10. With foraging, there is an element of exploration, actual consumption, and/or energy expenditure that are absent in the study design. Similarly, the main premise that seems to be utilized about the Stone-Geary utility framework is the idea that consumers have a baseline level of consumption for goods that needs to be met before the utility of that good is influenced by consumer preferences. The application of the Stone-Geary utility framework does seem to have a direct 1:1 correspondence with the present study design. Instead, based on the methods description, it seems that ‘need’ is manipulated based on the goal or target amount participants are told to aim for. If this is the case, it seems that there is a relationship between need and goal/goal setting. The description of need in the manuscript is difficult to grasp in the context of the present study. A more concrete connection to the present’ study methodology and connection to goals/goal setting could make this clearer. The paper cited below is an example of goal setting and risk-taking that seems to align well with the present study design.

Schiebener, J., Wegmann, E., Pawlikowski, M., & Brand, M. (2014). Effects of goals on decisions under risk conditions: Goals can help to make better choices, but relatively high goals increase risk-taking. Journal of Cognitive Psychology, 26(4), 473-485.

(2) Sample size & Power analysis: The sample size seems extremely small. Please state the effect sizes observed in Diederich et al. that were used in the power analysis. What hypotheses were the power analyses conducted for? What statistical analysis was the power analysis modeled off of (e.g., t-test, correlations, regression, etc)? While the within-subjects design is a strength of the study, a larger sample size or replication study would strengthen the conclusions drawn from study.

(3) “Finally, we included two response time limits: 1 second or 3 seconds for a given block of trials.” How was this time limit calculated? In other words, what is the scientific evidence to show that 1 and 3 seconds are appropriate time limits? Was there a pilot test to determine average response time, and time limits were adjusted from there? Or, something else? I can see that these are the time limits used in the authors’ previous paper (Diederich et al., 2020), but the evidence for these time limits is not specified in this previous paper either.

(4) Results: were outliers screened for in the Figure 3 analysis? Visually, it looks like there may be an outlier in the decisions about gains graph.

Minor:

• The idea of calling the initial amount of points a budget is a bit of a stretch. Budgeting with low-stakes points does not seem to have sufficient external validity to truly call it a budget.

• “Responses to a triple were collapsed for evaluation.” I don’t understand what this means.

• The level of detail in the Apparatus section isn’t essential to understanding the study given that figures are provided.

• In general, the methods are difficult to follow and could be improved with better organization and distinctive subheadings for each manipulation (frames, time limits, need levels, etc).

• Discussion: a conclusion paragraph would be helpful

6. PLOS authors have the option to publish the peer review history of their article (what does this mean?). If published, this will include your full peer review and any attached files.

Reviewer #1: No

Reviewer #2: No

---

## [Decision Letter · Decision Letter 1]

9 Mar 2022

Keep your budget together! Investigating determinants on risky decision-making about losses.

PONE-D-21-23927R1

Dear Dr. Wyszynski,

We’re pleased to inform you that your manuscript has been judged scientifically suitable for publication and will be formally accepted for publication once it meets all outstanding technical requirements.

I sent your manuscript back to one of the original reviewers and the reviewer felt all previous comments had been addressed.

Kind regards,

Darrell A. Worthy, Ph.D

Academic Editor

PLOS ONE

Additional Editor Comments (optional):

Reviewers' comments:

Reviewer's Responses to Questions

**Comments to the Author**

1. If the authors have adequately addressed your comments raised in a previous round of review and you feel that this manuscript is now acceptable for publication, you may indicate that here to bypass the “Comments to the Author” section, enter your conflict of interest statement in the “Confidential to Editor” section, and submit your "Accept" recommendation.

Reviewer #2: All comments have been addressed

2. Is the manuscript technically sound, and do the data support the conclusions?

Reviewer #2: Yes

3. Has the statistical analysis been performed appropriately and rigorously? 

Reviewer #2: Yes

4. Have the authors made all data underlying the findings in their manuscript fully available?

Reviewer #2: Yes

5. Is the manuscript presented in an intelligible fashion and written in standard English?

Reviewer #2: Yes

6. Review Comments to the Author

Reviewer #2: The authors have been significant changes that have improved the manuscript, including increasing the sample size from 28 to 44 participants, reporting the effect size used for sample size determination, reorganizing the method section, and adding a conclusion paragraph to the Discussion section.

7. PLOS authors have the option to publish the peer review history of their article (what does this mean?). If published, this will include your full peer review and any attached files.

Reviewer #2: No

---

## [Editor Report · Acceptance letter]

11 Mar 2022

PONE-D-21-23927R1 

Keep your budget together! Investigating determinants on risky decision-making about losses. 

Dear Dr. Wyszynski:

I'm pleased to inform you that your manuscript has been deemed suitable for publication in PLOS ONE. Congratulations! Your manuscript is now with our production department. 

Kind regards, 

on behalf of

Dr. Darrell A. Worthy 

Academic Editor

PLOS ONE